# Menstrual Cycle Disorders in Professional Female Rhythmic Gymnasts

**DOI:** 10.3390/ijerph16081470

**Published:** 2019-04-25

**Authors:** Mariola Czajkowska, Ryszard Plinta, Magdalena Rutkowska, Anna Brzęk, Violetta Skrzypulec-Plinta, Agnieszka Drosdzol-Cop

**Affiliations:** Chair of Woman’s Health, Medical University of Silesia, ul. Medyków 12, 40-752 Katowice, Poland; czajkowskamariola@op.pl (M.C.); ryszardplinta@wp.pl (R.P.); magdalenawodarska@poczta.fm (M.R.); aniabrzek@gmail.com (A.B.); skrzypulec-plinta@o2.pl (V.S.-P.)

**Keywords:** rhythmic gymnastics, menstruation disorders, premenstrual syndrome, premenstrual dysphoric disorder

## Abstract

The aim of this research was to compare menstrual cycles, menstrual disorders, and the prevalence of premenstrual syndrome (PMS)and premenstrual dysphoric disorder (PMDD) in professional female gymnasts and their peers who donot practice any sport, and to identify factors causing a predisposition to premenstrual tension syndrome and premenstrual dysphoric disorders in both groups. The prospective study involved apopulation of 85 girls. The study group consisted of 45 professional female gymnasts (15–17 years of age) who lived inthe territory of Silesia, in the southern area of Poland. The control group consisted of 40 girls of the same age who lived in the same area but did not professionally practice any sport. The research tools included a questionnaire, a daily diary of PMS symptoms, a daily diary of PMDD symptoms, and a premenstrual symptom screening tool (PSST). The study showed that intensive physical activity undertaken by girls before their first menstruation is a menarche-delaying factor andthat competitive sport promotes premenstrual syndrome and premenstrual dysphoric disorder. The risk factors for PMS and PMDD were also identified, andincluded alcohol and coffee consumption.

## 1. Introduction

The number of professional female athletes and the number of disciplines they practice have increased over the past few years. In accordance, so has the interest of researchers with respect tothe health consequences of this tendency [1].

In literature on the impact of physical activity on female physiology, authors unanimously emphasize that a significant percentage of women practicing various sport disciplines suffer from numerous menstrual cycle disorders. Important factors in their pathogenesis include young gynecological age, emotional or mental stress, a rapid increase in training load, low body weight, and eating disorders [1,2,3]. Long-term training periods and competitions impair ovarian activity, and this can manifest as luteal phase defects, irregular menstruation, or amenorrhea. These disorders are always accompanied by the lowered blood estradiol and progesterone levels during the luteal phase; however, the influence of sports activity on the level of estradiol and progesterone still needsadditional research [1,2,3,4,5,6,7].

Menstrual disorders caused by excessive physical effort often take form of a secondary amenorrhea and in some cases force women to give up their sport careers. They can also be caused by strong mental stress associated with practicing sport and can manifest withvarious mental, physical, and emotional symptoms during this time, including premenstrual syndrome (PMS) and premenstrual dysphoric disorder (PMDD) [4,5,6,7].

Premenstrual syndrome (PMS) is defined as cyclic somatic or psychopathological disorders associated with the existence of the corpus luteum and the influence of ovarian steroid metabolites on the level and activity of neurotransmitters. This condition can be considered physiological even though the severity of symptoms may be troublesome. Women of childbearing age in most of their menstrual cycles experience at least a few of the 150 symptoms of premenstrual syndrome. However, this syndrome is recognized according to very strict criteria.

Although premenstrual dysphoric disorder (PMDDD) is a recognized medical diagnosis, it is difficult to determine the exact boundaries between PMDD and normal, cyclic changes in the body related to premenstrual period due to a wide variety of symptoms including both mild physiological changes and clearly visible disorders of a pathological nature. The severity of symptoms, especially mental tension and lack of self-esteem, which are characteristic of PMDD, may cause inability to work and problems in coping with routine tasks. The occurrence of premenstrual dysphoric disorders may also be a risk factor fordeveloping depression in the future [6,7].

Disciplines with a high risk of all above menstrual disorders include aesthetic sports which require intensive training to achieve excellent flexibility, endurance, balance, motor coordination, rhythm, and dance skills. Rhythmic gymnastics is one of these sports; therefore the identification of menstrual disorders among female athletes practicing rhythmic gymnastics seemsessential to develop early intervention and prevention schemes and apply proper education of athletes and their parents, coaches, and sports doctors. Yet, thus far there has not been any research investigating the impact of physical effort made during competitive sports on the incidence of PMS and PMDD or menstrual disorders as such.

The aim of this research was to compare menstrual cycles, menstrual disorders, and the prevalence of PMS and PMDD in professional female gymnasts and their peers who donot practice any sports. The researchers also aimed to identify factors predisposing to premenstrual tension syndrome and premenstrual dysphoric disorders in both the study and control groups.

## 2. Materials and Methods

### 2.1. Study Population

The prospective study involved the population of 85 girls. Their characteristics are presented in the table below (Table 1).

The study group consisted of 45 young female athletes (15–17 years of age). They all lived in the territory of Silesia, in the south of Poland, and were members of sport clubs and/or students of sport schools.

The characteristics of the training sessions undertaken by the study group are presented in Table 2.

The surveyed girls practiced 8.24 ± 1.78 years, 6.04 ± 0.63 times a week, with each training session lasting 2.04 ± 0.29 h.

The control group consisted of 40 girls of the same age who lived in the same area but did not professionally practice any sport. They only took part in regular fitness lessons or occasionally engaged in other activities, but only as amateurs.

The study began only after it had been approved and accepted by the Bioethics Committee (KNW/022/KB/145/15). The participation in the study was entirely voluntary. The researchers also obtained the consent of the patient, and in case of minorsthe consent of their legal guardians according tothe Helsinki Declaration.

### 2.2. Research Methods

The research tools used in this study included a research questionnaire, PMS [8] and PMDD [9] symptom diaries, and a premenstrual symptom screening tool (PSST).

A research questionnaire contained questions about social and demographic status, family, the course of menstrual cycle, obstetrical and gynecological history, gynecological diseases, type of sport, the time and intensity of training, life style, everyday diet (daily observation), stimulants used, present health condition, current medications, and the occurrence of PMS and PMDD symptoms.

The daily diary of PMS symptoms was developed in accordance with The American College of Obstetricians and Gynecologists (ACOG) recommendations and ICD-10 diagnostic criteria.

The ACOG claims that premenstrual syndrome (PMS) cannot be diagnosed unless one of the six psychological symptoms or one of the four physical symptoms are observed. To diagnose PMS the symptoms must occur 5 days prior to the menstrual bleed in each of the two successive menstrual cycles (prospectively) and they must alleviate within 4 days after the onset of menstruation. Additionally, none of the symptoms can reappear until at least the 13th day of the menstrual cycle and before the pre-ovulatory phase. One must also exclude the use of pharmacotherapy, hormone intake, and social or economic impairments or dysfunctions. The subjects of the study were also asked about the frequency of drinking coffee, alcohol, using drugs, or following specific diets such as protective, no-residue, light, low calorie, high protein, vegetarian, or starvation diets.

A daily diary of premenstrual dysphoric disorder (PMDD) symptoms has been developed according to DSM-V diagnostic criteria of the American Psychiatric Association (APA). They are someof the most complex diagnostic criteria. The diagnosis of premenstrual dysphoric disorders must be confirmed by the occurrence of at least 5 out of 11 symptoms, where at least one of them should indicate depression, anxiety, excitability or emotionally labile disorders. These symptoms must impair everyday life, affecting work, social life, and/or emotional relationships. At the same time the presence of another chronic disease must be excluded. The symptoms should also occur cyclically and appear each time in the luteal phase and disappear in the follicular phase for at least two consecutive menstrual cycles, which must be confirmed by daily entries in the diary. PMDD can also be recognized if the above criteria are met in most of the menstrual cycles during the previous calendar year.

To verify the occurrence of premenstrual syndrome and premenstrual dysphoric disorders the researchers also used a premenstrual symptoms screening tool (PSST). PSST classification allowedus to identify the following scenarios: PMS with no PMDD; heavy PMS; moderate PMS; PMDD with heavy PMS; and PMDD with moderate PMS.

### 2.3. Statistical Analysis

Parameters of normal distribution were presented as mean and standard deviations (X ± SD), and differences between the groups were evaluated with Student’s *t*-test. Due to the population of groups, normal distribution was examined with the Kolmogorov–Smirnov test. Advanced parameters with non-normal distribution were presented as a median (Me) and differences between groups were verified using Wilcoxon (Z) and Mann-Whitney (U) tests. Quality parameters were presented as a percentage and the differences between the groups were checked by means of a chi-squared test (the Yates correction was applied) and stratum weight. The correlations were evaluated with R-Pearson and R-Spearman tests (the Fisher’s exact test was used). The statistically significant level was *p* < 0.05. Statistical analysis was carried out using StatisticaSoftware v.13 program (Statistica v.10, StatSoft Polska, Krakow, Poland).

## 3. Results

The research showed that the incidence of pain during menstruation was not statistically different in two groups and averaged according to a visual analogue scale (VAS)score of 1.95 ± 0.47 (95 CI: 1.81–2.09) in the study group and 1.85 ± 0.47 (95 CI: 1.71–1.98) in the control group. AVAS is a measurement instrument that tries to measure a characteristic or attitude that is believed to range across a continuum of values and cannot easily be directly measured, with 0 denoting no pain and 10 denoting the worst pain imaginable for the purpose of this particular study. It was found that 10.26% of female athletes complained about pain lasting from the beginning to the end of menstruation. None of the girls from the control group complained about such long periods of pain. It was also found that 11.11% of female gymnasts started sexual intercourse at the age of 17, while 17.50% of girls in the control group started sexual activities at the ages of 16 and 17. The examination of menstrualcycle abnormalities showed longer breaks between menstruations only among girls from the study group. Sixty percent of study group participants suffered from hypermenorrhea, as compared to their peers from the control group (*p* < 0.0002) (Table 3).

For an in-depth evaluation of the variables, the researchers analyzed menstrual complaints of respondents from both groups. This analysis showed that there were no statistically significant differences in the number of symptoms reported between the groups (Figure 1).

Then, on the basis of APA and ACOG guidelines, the prevalence of PMDD and PMS in respondents from both groups was assessed. The groups were homogeneous (*p* > 0.05) in terms of the prevalence of both disorders and severity of premenstrual symptoms (Table 4).

No correlation between the prevalence of PMS and PMDD and the age of respondents or the age of their first menstruation was observed in any of the groups. The years of practice, its frequency, or duration of training sessions did not affect the prevalence of PMS and PMDD in gymnasts (all *p* > 0.05) (Table 5). PMS and PMDD occurrence in gymnasts was, however, influenced by the amount of coffee and alcohol they drank (Table 5).

## 4. Discussion

### 4.1. Influence of Sport on Menstrual Cycles

Any sport that requires a high level of performance exposes young athletes to severe physical, mental, and emotional stress, and central mechanisms activating hypothalamo–pituitary–adrenal axis may cause a number of disorders [5,10,11,12]. Young female athletes are additionally exposed to a constellation of menstrual cycle disorders such as hypermenorrhoea, longer breaks between menstrual bleeds, and delays in menarche as well as PMS and PMDD symptoms. Disciplines with a high risk for menstrual disorders include aesthetic sports such as rhythmic gymnastics, and hence it is important to ascertain if menstrual disorders and the prevalence of PMS and PMDD differ between professional female gymnasts and their peers who do not practice any sport. There is a need to identify factors causing a predisposition to premenstrual tension syndrome and premenstrual dysphoric disorders in both groups.

The study showed a number of menstrual disorders in the study group, including longer breaks between menstrual bleeds and hypermenorrhoea in 60% of female athletes (*p* < 0.0002). A statistically significant difference (*p* < 0.05) in the frequency of spotting in the study group was also observed. The incidence of pain during menstrual bleed averaged (according to VAS) 1.95 ± 0.47 (95 CI: 1.81–2.09) in the study group and 1.85 ± 0.47 (95 CI: 1.71–1.98) in the control group. It was found that 10.26% of female athletes complained about pain lasting from the beginning to the end of menstrual bleed, while none of the girls from the control group reported such long periods of pain.

The study also confirmed the results of other studies [4,10,11,12,13,14,15,16], which showed that intensive physical activity is a menarche-delaying factor. In this research an average age of menarche was 13.02 ± 1.03 years in the study group and 12.82 ± 1.22 in the control group, which reflects the norms specific to Central Europe. Both in this study and indicated literature the age of menarche was determined retrospectively by means of questionnaires. However, it should be noted that a significant limitation to the studies of female athletes with high sport achievements is the small number of groups [4,5,17,18,19,20,21].

These findingsseem to be of great cognitive importance, with practical implications forpediatric/adolescentgynecology and sports medicine to ensure early intervention and prevention based on the education of athletes, their parents, coaches, and sports doctors.

### 4.2. Risk Factors for PMS and PMDD

The study also showed that PMDD frequency did not significantly differ between the groups (13.13% versus 5.0%) and nor did PMS (48.89% versus 32.5%). PSST differentiated PMS and PMDD into mild and severe. Mild PMS was diagnosed in a significant percentage of women in both the study group (31.11%) and control group (27.50%). However, the differences between the groups were statistically insignificant. The study did not show any significant PMDD and PMS dependence on girls’ age or age of menarche in both analyzed groups. In the group of athletes there was also no dependence on the time they practiced, the number of trainings they had per week or their diet.

However, a significant influence of the amount of coffee and alcohol drunk on the incidence of PMS and PMDD in female athletes was noted, which confirms the results of studies by Kantanista et al. [22]. In addition, the researchers point to greater coffee and alcohol consumption among less educated women, suggesting that education might determine the awareness of health-promoting behaviors, which again confirms findings by other researchers [23,24,25,26,27,28].

So far only the influence of physical effort on the intensity of PMS symptoms has been investigated (Melin et al. [14]) but no other studies have been attempted to verify the impact of physical effort made during competitive sports on the incidence and frequency of both PMS and PMDD according to APA and ACOG criteria. In this respect, the combination of daily prospective assessment of PMS and PMDD symptoms, the use of the screening tool for premenstrual symptoms (PSST), and the division of the subjects into non-athletes and athletes practicing one discipline presents a truly innovative approach.

There are of course a number of limitations to this study. Firstly, the small size of the study group makes it impossible to reliably generalize the results to the entire population of adolescent rhythmic gymnasts. Secondly, the authors did not evaluate the influence of self-esteem, expectations about menstruation, state and trait stress level, or the level of identification with traditional social roles on the risk of PMS/PMDD. The role of these factors has been reported in other studies, but these are not Polish. Thirdly, no clinical/psychiatric diagnosis of mental health problems was performed in the study population. Unfortunately, young women with mental health issues might present symptoms overlapping with the symptoms of PMS. The limitations of the current study also include its cross-sectional design and cohort size. Therefore, further research is needed to understand the individual and social impact of the problemon individual professional female rhythmic gymnasts and on the quality of their life. It is also worth considering expanding current research to other competitive sports.

## 5. Conclusions

Alcohol and coffee consumption are important risk factors for premenstrual tension syndrome and premenstrual dysphoric disorders in female athletes. Competitive sport promotes premenstrual syndrome and premenstrual dysphoric disorder, and regular training undertaken by girls before their first menstruation is a menarche-delaying factor.

## Figures and Tables

**Figure 1 ijerph-16-01470-f001:**
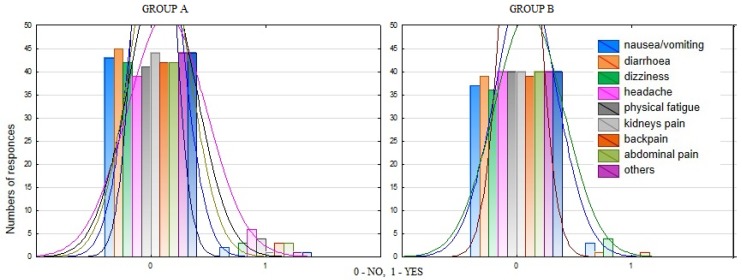
The distribution of the test material in terms of the incidence of other complaints during menstrual bleed.

**Table 1 ijerph-16-01470-t001:** Characteristics of both groups in terms of age, Body Mass Index (BMI) and age of first menstruation.

Variables	Study Group (*n* = 45)				Control Group (*n* = 40)			
	X	SD	Me	95% CI	X	SD	Me	95% CI
Age (years)	16.28	0.84	17	16.03–16.54	16.93	1.02	17	16.59–17.25
Menarche (age)	13.02	1.03	13	12.71–13.33	12.82	1.22	13	12.43–13.21

Abbreviations—X: average; SD: standard deviations; Me: median; 95% CI: confidence interval.

**Table 2 ijerph-16-01470-t002:** Characteristics of the study group in terms of frequency and duration of their training sessions.

Variables	Study Group			
	X	SD	Me	95% CI
Years of practice	8.24	1.78	8	7.7–8.78
Frequency per week (days)	6.04	0.63	6	5.85–6.23
Duration of one training session (hours)	2.04	0.29	2	1.95–2.13

**Table 3 ijerph-16-01470-t003:** Characteristics of the groups including gynecological history.

Variables	Study Group		Control Group		*p* ^a^
	*N*	%	*N*	%	
Regularity of menstruation (yes)	33	73.33	25	62.5	0.28
Occurrence of menstruation (yes)	45	100	40	100	-
Every 24 days	6	13.33	6	15	0.82
Every 25–31 days	32	71.4	23	57.5	0.18
More than 31 days	7	15.56	11	27.5	0.17
Longer breaks between menstruations (yes):	17	37.78	20	25	0.25
≤3 months	9	52.94	1	10	**
3–6 months	8	47.06	8	80	-
≤6 months	-	-	-	-	-
Hypermenorrhoea (yes)	28	62.22	9	22.5	***
Pain during menstruation (yes)	39	86.67	35	87.5	0.9
Spotting between menstruations (yes)	9	20	2	5	*
Contraceptive pills (yes)	-	-	-	-	

^a^ stratum weight was used, * *p* < 0.05, ** *p* < 0.01, *** *p* < 0.001.

**Table 4 ijerph-16-01470-t004:** Characteristics of the groups in terms of PMS and PMDD prevalence and screening PSST.

Variables	Study Group		Control Group		*p* ^a^
	*N*	%	*N*	%	
PMS (YES)	22	48.89	13.	32.5	0.12
PMDD (YES)	6	13.13	2	5	0.19
PSST					
PMS with no PMDD	23	51.11	27	67.5	0.12
Heavy PMS	2	4.44	-	-	0.18
Moderate PMS	14	31.11	11	27.5	0.71
PMDD and heavy PMS	2	4.44	1	2.5	0.63
PMDD and moderate PMS	4	8.89	1	2.5	0.21

Abbreviations—PMS: premenstrual syndrome; PMDD: premenstrual dysphoric disorder; PSST: premenstrual symptoms screening tool; ^a^ stratum weight was used, *p* < 0.05, * *p* < 0.01, ** *p* < 0.001.

**Table 5 ijerph-16-01470-t005:** Factors influencing PMS and PMDD in both groups according to R-Spearman test.

Variables		Study Group		Control Group	
		R	*p*-Value	R	*p*-Value
PMS	Age	0.24	0.11	−0.13	0.41
	Menarche	−0.08	0.56	0.2	0.13
	Years of practice	0.16	0.28	-	-
	Training frequency	0.22	0.14	-	-
	Duration of one training	0.15	0.31	-	-
	Regularity of meals	0.03	0.83	0.13	0.39
	Diet	0.06	0.68	0.17	0.28
	Amount of coffee drunk	0.62 *	0.009	−0.1	0.65
	Frequency of alcohol consumption ^a^	0.34 *	0.021	0.03	0.81
PMDD	Age	0.008	0.96	0.19	0.23
	Menarche	−0.09	0.55	−0.06	0.7
	Years of practice	−0.13	0.38	-	-
	Training frequency	−0.12	0.4	-	-
	Duration of one training	−0.06	0.69	-	-
	Regularity of meals		0.99	−0.08	0.59
	Following specific diet	0.04	0.81	0.07	0.66
	Amount of coffee drunk	0.06	0.8	0.11	0.61
	Frequency of alcohol consumption ^a^	0.25	0.09	0.04	0.78

* *p* < 0.05. PMS: premenstrual syndrome; PMDD: premenstrual dysphoric disorder. ^a^ the frequency of alcohol consumption: not at all, rarely (during celebrations as a toast), often (during parties with friends).

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
