# Peer review of "Menstrual Cycle Disorders in Professional Female Rhythmic Gymnasts"

_ijerph, 2019, doi:10.3390/ijerph16081470_

Round 1

Reviewer 1 Report

The study provides new insights into menstruation related phenomena in female gymnasts in Poland. The study appears well constructed and novel.  There are some improvements needed to the writing style and presentation of results and the discussion would benefit from a stronger structure providing clarity over the key findings and the implications of these findings.  There is no statement of the aims in the introduction. No practical recommendations or clinical implications are provided in the discussion.  It is interesting that both sport related factors and lifestyle factors contribute (coffee and alcohol) to menstruation symptoms and this is worthy of further discussion.  A limitation that has been ignored in the discussion is that this is a cross sectional study providing associations and it does not determine cause and effect – this could also be discussed.   

Specific comments: 

Abstract 

Line 11: define terms

Line 14: the age data is typically placed in parentheses after the sample size

Line 16: remove ‘self-designed’

Line 18: use 1 decimal place for mean and SD age data

Line 19: reword ‘it was also proved that…’ since so far the only thing reported is the age of the participants. 

Line 20: remove ‘The research also showed that’

Line 22: define VAS

Line 24: define APA and ACOG

Line 25: be consistent with decimal places reported for % values

Line 26: insert ‘a’ between PMDD and Premenstrual

Line 27: what it ‘it’?

Line 28: remove ‘the conducted research allows for the conclusion’ and replace with ‘It was concluded’

Line 29: ‘trainings’ should be ‘training’

Line 30: Consider moving the last statement further up the abstract to the results section. 

Intro 

The introduction is lacking an explanation of the underlying physiology that causes symptoms, i.e. ovarian hormone fluctuations. 

There is are no specific aims of the study defined in the introduction. 

Line 36: Are any data available to support the first statement – could any specific examples be included? E.g. What are the data like in gymnastics, of most relevance to this study?

Line 39: Consider ‘female physiology’ rather than ‘women’s bodies’ 

Line 41: Lack of consistency with terminology: e.g. menstrual cycle vs. menstruation cycle vs. monthly cycle

Line 47: reword or remove: ‘however one cannot exclude the possibility that it affects people with primary personality disorders’.  I would argue that this is contentious and not relevant to the present study. 

Line 49-51: please provide a reference to support these statements.  Are these athletes and if not, how does it differ?

Line 56: replace menstruation with menstrual

Line 58: ‘official diagnostic unit’ – Could better terminology be used here? E.g. recognised medical diagnosis.

Line 63:  ‘household chores’ should be avoided due to the connotations with stereotypical female roles.  Suggest ‘routine tasks’ as an alternative.

Line 66: remove ‘as a conclusion’

Materials and methods

Please provide references for ACOG and APA documented cited.

Table 1: Include sample sizes in the table. 

Lines 74 – 76:  They are not professional unless they are paid to be gymnasts. 

Lines 93 – 95: Repetition of ‘professional’ numerous times.  A single explanation of the training and level of these gymnasts is sufficient. 

Line 98:  Suggest combining tables 1 and 2 to clearly show the differences between the groups. 

Line 109:  It would be helpful if these could be included as supplementary materials, or the source referenced. 

Line 113: ‘trainings’ should be ‘training’

Line 119:  replace ‘a menstruation’ with ‘a menstrual bleed’

Lines 115 and 114: Should be the daily diary wasdeveloped

Lines 116 and 124: define terms

Line 129: replace ‘exacerbation’ with ‘presence’

Line 145: spelling: software

Results: 

Line 148: Please explain the VAS scores – are they 0-10 scales? Define the extremes, e.g. 0 = no pain? 10 = extreme pain? These VAS scores are not explained in the methods

Line 154 and Table 3: how were heavy periods defined? 

Figure 1:  There are a number of problems with this figure.  If the y-axis scales were the same for Group A and Group B it would be easier to see the differences.  The distribution curves are not needed unless some explanation can be provided for their meaning and purpose. The text is poor quality. Change y-axis label to ‘number of responses’

Table 4: include PSST in the title.  Greater explanation of what the variables are is required, e.g. ‘No’.  Suggest using footnotes to explain all variables and abbreviations in all tables and figures.

Table 5: it would be helpful to see the actual P values, given there are some strong correlations. Left-justify ‘frequency of alcohol consumption’. Some variables need an explanation, e.g. ‘Diet’

Discussion: 

Broadly, the English style could be improved through the discussion.  Furthermore, it might benefit from some restructuring as follows: Statement of the key findings and the strengths of the study; Context of previous literature; Limitations; Recommendations; Conclusions.  

Line 182:  Suggest replacing ‘obtain clear results determining’ with ‘determine’ 

Lines 184 – 188:  This is poorly written and needs rewording to provide clarity on the differences between gymnastics vs. the control group.

Line 189: This is a single sentence paragraph. Maybe it should be subheading, or worked into a full paragraph referring to the data that supports this statement. 

Line 198: Suggest replacing ‘This way an’ with ‘therefore’.

Line 215: This a statement of results and needs a discussion comment – perhaps combine with the following paragraph? 

Conclusion: 

Line 264: The first statement is not clear.  It is suggested that the methods demonstrated in this study are useful for screening for risk factors?   

Author Response

Dear Reviewer 1,

Thank you very much for such in-depth and detailed analysis and verification of our work. We read all your comments, suggestions and remarks very thoroughly  and we believe they are all well substantiated and their implementation willdefinitely make the text more clear, coherent and reader friendly.

Therefore, following your suggestions we haveexpanded all abbreviations, corrected all decimal places, added missing references, solved all inconsistencies, corrected all Tables and Figures (apart from Table1 and Table 2. In this case we appreciate the idea of combining two tables into one, however, we believe they should stay as they are because the control group includes non-athletes and it’s not possible to include information about years of practice or frequency of training), added specific aims to the introduction section, changed the whole Discussion paragraph according to the suggested scheme and added the context of available literature, missing strengths and limitations of the study, recommendations, practical implications and stressed the need for further research.

We do believe that this new version of the paper is better structured with the rationale and significance better expressed. And we hope that with Your help the publication will offer other scientists an interesting insight into menstruation related problems in female gymnasts. Thank you very much!

Yours sincerely,

Mariola Czajkowska , Ryszard Plinta , Magdalena Rutkowska , Anna Brzęk , Violetta Skrzypulec-Plinta , Agnieszka Drosdzol–Cop

Reviewer 2 Report

Thank you for the opportunity to review your work.  Overall, the methods and findings are well-articulated, but the rationale and significance for this study is not well expressed, nor are the limitations of this study in the discussion section.  Specific feedback is listed as follows:

ABSTRACT

·      Ln 11, spell out PMS and PMDD in abstract before using acronym

·      Ln 22, spell out VAS before using acronym

·      Ln 25-27, Sentence reads odd, please insert commas where needed

INTRODUCTION

·      Some grammar and wordsmithing is needed throughout.

·      Ln 52, PMS and PMDD need to be spelled out here

·      Ln 58, “diagnostic unit” reads odd to me.  Do you mean health condition?

·      Ln 66, remove “… Polish or foreign authors…”  Instead consider replacing with a thought about how this literature has not been researched and data is not found currently on this topic in the literature.

·      No justification is given and no prior research is cited as to why the focus is on rhythmic gymnastics.  This needs to be addressed in the introduction.  Why this population?  Are they at particular risk from a physical and psychological context? 

·      The rationale for performing this study is invalid and unsatisfactory.  It is not good enough to perform a study just because it has not yet been done.  What is the importance of this study?  What would findings contribute to from a public health/sport standpoint?  Readers need to feel (at the end of the introduction) that this study is really important to perform.  At this point, I don’t feel that at all.

METHODS

·      Ln 77-92, I feel these should be located in the introduction section

·      Ln 103, was control group physical activity captured?  Important to control for these if possible

·      Methods section presented clearly

DISCUSSION

·      Ln 189-192, sentence should be rewritten.  As it stands, it is difficult to make sense of

·      Ln 193, should state the differences observed between studies “likely” result form type of discipline

·      Ln 202, by “arithmetic” do you mean, “calculated”?

·      Ln 223, remove “it also confirmed”

·      Ln 231, a comma must be inserted into this sentence

·      No discussion of how findings for rhythmic gymnastics may differ from those for other competitive sports

·      Limitations and strengths of this study have not been addressed or listed properly in this section

Author Response

Dear Reviewer 2,

Thank you very much for such in-depth and detailed analysis and verification of our work. We read all your comments, suggestions and remarks very thoroughly  and we believe they are all well substantiated and their implementation willdefinitely make the text more clear, coherent and reader friendly.

Therefore, following your suggestions we haveexpanded all abbreviations, rephrased odd sentences and terminology, added specific aims to the introduction section, explained the relation between rhythmic gymnastics and other competitive sports, changed the whole Discussion paragraph according to the scheme:Statement of the key findings and the strengths of the study in the context of previous literature à Limitations to the study à Recommendations. We also mentioned practical implications and stressed the need for further research.

We do believe that this new version of the paper is better structured with the rationale and significance better expressed. And we hope that with Your help the publication will offer other scientists an interesting insight into menstruation related problems in female gymnasts. Thank you very much!

Yours sincerely,

Mariola Czajkowska , Ryszard Plinta , Magdalena Rutkowska , Anna Brzęk , Violetta Skrzypulec-Plinta , Agnieszka Drosdzol–Cop

Round 2

Reviewer 2 Report

N/A